# RECTIFIED FACTOR NETWORKS FOR BICLUSTERING

**Djork-Arné Clevert, Thomas Unterthiner, & Sepp Hochreiter**
Institute of Bioinformatics
Johannes Kepler University, Linz, Austria
{okko,unterthiner,hochreit}@bioinf.jku.at

## ABSTRACT

Biclustering is evolving into one of the major tools for analyzing large datasets given as matrix of samples times features. Biclustering has several noteworthy applications and has been successfully applied in life sciences and e-commerce for drug design and recommender systems, respectively.

FABIA is one of the most successful biclustering methods and is used by companies like Bayer, Janssen, or Zalando. FABIA is a generative model that represents each bicluster by two sparse membership vectors: one for the samples and one for the features. However, FABIA is restricted to about 20 code units because of the high computational complexity of computing the posterior. Furthermore, code units are sometimes insufficiently decorrelated. Sample membership is difficult to determine because vectors do not have exact zero entries and can have both large positive and large negative values.

We propose to use the recently introduced unsupervised Deep Learning approach Rectified Factor Networks (RFNs) to overcome the drawbacks of existing biclustering methods. RFNs efficiently construct very sparse, non-linear, high-dimensional representations of the input via their posterior means. RFN learning is a generalized alternating minimization algorithm based on the posterior regularization method which enforces non-negative and normalized posterior means. Each code unit represents a bicluster, where samples for which the code unit is active belong to the bicluster and features that have activating weights to the code unit belong to the bicluster.

On 400 benchmark datasets with artificially implanted biclusters, RFN significantly outperformed 13 other biclustering competitors including FABIA. In biclustering experiments on three gene expression datasets with known clusters that were determined by separate measurements, RFN biclustering was two times significantly better than the other 13 methods and once on second place. On data of the 1000 Genomes Project, RFN could identify DNA segments which indicate, that interbreeding with other hominins starting already before ancestors of modern humans left Africa.

## 1 INTRODUCTION

Biclustering is widely-used in statistics (A. Kasim & Talloen, 2016), and recently it also became popular in the machine learning community (O´ Connor & Feizi, 2014; Lee et al., 2015; Kolar et al., 2011), e.g., for analyzing large dyadic data given in matrix form, where one dimension are the samples and the other the features. A matrix entry is a feature value for the according sample. A *bicluster* is a pair of a sample set and a feature set for which the samples are similar to each other on the features and vice versa. Biclustering simultaneously clusters rows and columns of a matrix. In particular, it clusters row elements that are similar to each other on a subset of column elements. In contrast to standard clustering, the samples of a bicluster are only similar to each other on a subset of features. Furthermore, a sample may belong to different biclusters or to no bicluster at all. Thus, biclusters can overlap in both dimensions. For example, in drug design biclusters are compounds which activate the same gene module and thereby indicate a side effect. In this example different chemical compounds are added to a cell line and the gene expression is measured (Verbist et al., 2015). If multiple pathways are active in a sample, it belongs to different biclusters and may

have different side effects. In e-commerce often matrices of costumers times products are available, where an entry indicates whether a customer bought the product or not. Biclusters are costumers which buy the same subset of products. In a collaboration with the internet retailer Zalando the biclusters revealed outfits which were created by customers which selected certain clothes for a particular outfit.

FABIA (factor analysis for bicluster acquisition, (Hochreiter et al., 2010)) evolved into one of the most successful biclustering methods. A detailed comparison has shown FABIA's superiority over existing biclustering methods both on simulated data and real-world gene expression data (Hochreiter et al., 2010). In particular FABIA outperformed non-negative matrix factorization with sparseness constraints and state-of-the-art biclustering methods. It has been applied to genomics, where it identified in gene expression data task-relevant biological modules (Xiong et al., 2014). In the large drug design project QSTAR, FABIA was used to extract biclusters from a data matrix that contains bioactivity measurements across compounds (Verbist et al., 2015). Due to its successes, FABIA has become part of the standard microarray data processing pipeline at the pharmaceutical company Janssen Pharmaceuticals. FABIA has been applied to genetics, where it has been used to identify DNA regions that are identical by descent in different individuals. These individuals inherited an IBD region from a common ancestor (Hochreiter, 2013; Povysil & Hochreiter, 2014). FABIA is a generative model that enforces sparse codes (Hochreiter et al., 2010) and, thereby, detects biclusters. Sparseness of code units and parameters is essential for FABIA to find biclusters, since only few samples and few features belong to a bicluster. Each FABIA bicluster is represented by two membership vectors: one for the samples and one for the features. These membership vectors are both sparse since only few samples and only few features belong to the bicluster.

However, FABIA has shortcomings, too. A disadvantage of FABIA is that it is only feasible with about 20 code units (the biclusters) because of the high computational complexity which depends cubically on the number of biclusters, i.e. the code units. If less code units were used, only the large and common input structures would be detected, thereby, occluding the small and rare ones. Another shortcoming of FABIA is that units are insufficiently decorrelated and, therefore, multiple units may encode the same event or part of it. A third shortcoming of FABIA is that the membership vectors do not have exact zero entries, that is the membership is continuous and a threshold have to be determined. This threshold is difficult to adjust. A forth shortcoming is that biclusters can have large positive but also large negative members of samples (that is positive or negative code values). In this case it is not clear whether the positive pattern or the negative pattern has been recognized.

Rectified Factor Networks (RFNs; (Clevert et al., 2015)) RFNs overcome the shortcomings of FABIA. The first shortcoming of only few code units is avoided by extending FABIA to thousands of code units. RFNs introduce rectified units to FABIA's posterior distribution and, thereby, allow for fast computations on GPUs. They are the first methods which apply rectification to the posterior distribution of factor analysis and matrix factorization, though rectification it is well established in Deep Learning by rectified linear units (ReLUs). RFNs transfer the methods for rectification from the neural network field to latent variable models. Addressing the second shortcoming of FABIA, RFNs achieve decorrelation by increasing the sparsity of the code units using dropout from field of Deep Learning. RFNs also address the third FABIA shortcoming, since the rectified posterior means yield exact zero values. Therefore, memberships to biclusters are readily obtained by values that are not zero. Since RFNs only have non-negative code units, the problem of separating the negative from the positive pattern disappears.

## 2    IDENTIFYING BICLUSTERS BY RECTIFIED FACTOR NETWORKS

### 2.1    RECTIFIED FACTOR NETWORKS

We propose to use the recently introduced Rectified Factor Networks (RFNs; (Clevert et al., 2015)) for biclustering to overcome the drawbacks of the FABIA model. The factor analysis model and the construction of a bicluster matrix are depicted in Fig. 1. RFNs efficiently construct very sparse, non-linear, high-dimensional representations of the input. RFN models identify rare and small events in the input, have a low interference between code units, have a small reconstruction error, and explain the data covariance structure.

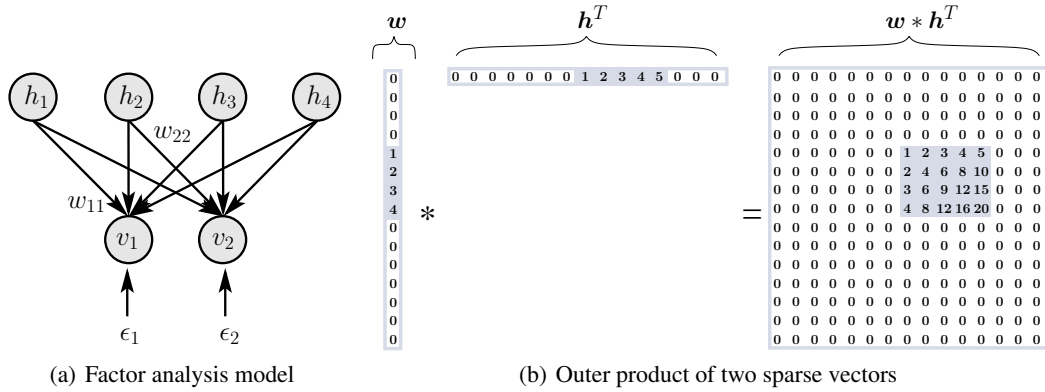

(a) Factor analysis model          (b) Outer product of two sparse vectors

Figure 1: Left: Factor analysis model: hidden units (factors) $h$, visible units $v$, weight matrix $W$, noise $\epsilon$. Right: The outer product $w\, h^T$ of two sparse vectors results in a matrix with a bicluster. Note that the non-zero entries in the vectors are adjacent to each other for visualization purposes only.

RFN learning is a generalized alternating minimization algorithm derived from the posterior regularization method which enforces non-negative and normalized posterior means. These posterior means are the code of the input data. The RFN code can be computed very efficiently. For non-Gaussian priors, the computation of the posterior mean of a new input requires either to numerically solve an integral or to iteratively update variational parameters. In contrast, for Gaussian priors the posterior mean is the product between the input and a matrix that is independent of the input. RFNs use a rectified Gaussian posterior, therefore, they have the speed of Gaussian posteriors but lead to sparse codes via rectification. RFNs are implemented on GPUs.

The RFN model is a factor analysis model

$$v \; = \; Wh \; + \; \epsilon \, , \tag{1}$$

which extracts the *covariance structure* of the data. The prior $h \sim \mathcal{N}\left(0, I\right)$ of the hidden units (factors) $h \in \mathbb{R}^l$ and the noise $\epsilon \sim \mathcal{N}\left(0, \Psi\right)$ of visible units (observations) $v \in \mathbb{R}^m$ are independent. The model parameters are the weight (factor loading) matrix $W \in \mathbb{R}^{m \times l}$ and the noise covariance matrix $\Psi \in \mathbb{R}^{m \times m}$.

RFN models are selected via the posterior regularization method (Ganchev et al., 2010). For data $\{v\} = \{v_1, \ldots, v_n\}$, it maximizes the objective $\mathcal{F}$:

$$\mathcal{F} = \frac{1}{n} \sum_{i=1}^{n} \log p(v_i) - \frac{1}{n} \sum_{i=1}^{n} D_{\mathrm{KL}}(Q(h_i \mid v_i) \parallel p(h_i \mid v_i)), \tag{2}$$

where $D_{\mathrm{KL}}$ is the Kullback-Leibler distance. Maximizing $\mathcal{F}$ achieves two goals simultaneously: (1) extracting desired structures and information from the data as imposed by the generative model and (2) ensuring sparse codes via $Q$ from the set of rectified Gaussians.

For Gaussian posterior distributions, and mean-centered data $\{v\} = \{v_1, \ldots, v_n\}$, the posterior $p(h_i \mid v_i)$ is Gaussian with mean vector $(\mu_p)_i$ and covariance matrix $\Sigma_p$:

$$(\mu_p)_i \; = \; \left(I \; + \; W^T \Psi^{-1} W\right)^{-1} W^T \Psi^{-1} \, v_i \quad , \quad \Sigma_p \; = \; \left(I \; + \; W^T \Psi^{-1} W\right)^{-1} . \tag{3}$$

For rectified Gaussian posterior distributions, $\Sigma_p$ remains as in the Gaussian case, but minimizing the second $D_{\mathrm{KL}}$ of Eq. (2) leads to constrained optimization problem (see Clevert et al. (2015))

$$\min_{\mu_i} \; \frac{1}{n} \sum_{i=1}^{n} (\mu_i \; - \; (\mu_p)_i)^T \, \Sigma_p^{-1} \, (\mu_i \; - \; (\mu_p)_i)$$

$$\text{s.t.} \quad \forall_i : \, \mu_i \; \geq \; 0 \; , \; \forall_j : \frac{1}{n} \sum_{i=1}^{n} \mu_{ij}^2 \; = \; 1 \, , \tag{4}$$

where "$\geq$" is component-wise. In the E-step of the *generalized alternating minimization* algorithm (Ganchev et al., 2010), which is used for RFN model selection, we only perform a step of the *gradient projection algorithm* (Bertsekas, 1976; Kelley, 1999), in particular a step of the *projected Newton method* for solving Eq. (4) (Clevert et al., 2015). Therefore, RFN model selection is extremely efficient but still guarantees the correct solution.

## 2.2 RFN Biclustering

For a RFN model, each code unit represents a bicluster, where samples, for which the code unit is active, belong to the bicluster. On the other hand features that activates the code unit belong to the bicluster, too. The vector of activations of a unit across all samples is the sample membership vector. The weight vector which activates the unit is the feature membership vector. The un-constraint posterior mean vector is computed by multiplying the input with a matrix according to Eq. (3). The constraint posterior of a code unit is obtained by multiplying the input by a vector and subsequently rectifying and normalizing the code unit (Clevert et al., 2015).

To keep feature membership vector sparse, we introduce a *Laplace prior on the parameters*. Therefore only few features contribute to activating a code unit, that is, only few features belong to a bicluster. Sparse weights $\boldsymbol{W}_i$ are achieved by a component-wise independent *Laplace* prior for the weights:

$$p(\boldsymbol{W}_i) \;=\; \left(\tfrac{1}{\sqrt{2}}\right)^n \prod_{k=1}^{n} e^{-\sqrt{2}\,|\boldsymbol{W}_{ki}|} \tag{5}$$

The weight update for RFN (Laplace prior on the weights) is

$$\boldsymbol{W} \;=\; \boldsymbol{W} \;+\; \eta \left(\boldsymbol{U}\,\boldsymbol{S}^{-1} \;-\; \boldsymbol{W}\right) - \alpha \, \mathrm{sign}(\boldsymbol{W}) \,. \tag{6}$$

Whereby the sparseness of the weight matrix can be controlled by the hyper-parameter $\alpha$ and $\boldsymbol{U}$ and $\boldsymbol{S}$ are defined as $\boldsymbol{U} = \frac{1}{n} \sum_{i=1}^{n} \boldsymbol{v}_i \boldsymbol{\mu}_i^T$ and $\boldsymbol{S} = \frac{1}{n} \sum_{i=1}^{n} \boldsymbol{\mu}_i \boldsymbol{\mu}_i^T + \boldsymbol{\Sigma}$, respectively. In order to enforce more sparseness of the sample membership vectors, we introduce *dropout* of code units. Dropout means that during training some code units are set to zero at the same time as they get rectified. Dropout avoids co-adaptation of code units and reduces correlation of code units — a problem of FABIA which is solved.

RFN biclustering does not require a threshold for determining sample memberships to a bicluster since rectification sets code units to zero. Further crosstalk between biclusters via mixing up negative and positive memberships is avoided, therefore spurious biclusters do less often appear.

## 3 Experiments

In this section, we will present numerical results on multiple synthetic and real data sets to verify the performance of our RFN biclustering algorithm, and compare it with various other biclustering methods.

## 3.1 Methods Compared

To assess the performance of rectified factor networks (RFNs) as unsupervised biclustering methods, we compare the following 14 biclustering methods:

(1) **RFN**: rectified factor networks (Clevert et al., 2015), (2) **FABIA**: factor analysis with Laplace prior on the hidden units (Hochreiter et al., 2010; Hochreiter, 2013), (3) **FABIAS**: factor analysis with sparseness projection (Hochreiter et al., 2010), (4) **MFSC**: matrix factorization with sparseness constraints (Hoyer, 2004), (5) **plaid**: plaid model (Lazzeroni & Owen, 2002; T. Chekouo & Raffelsberger, 2015), (6) **ISA**: iterative signature algorithm (Ihmels et al., 2004), (7) **OPSM**: order-preserving sub-matrices (Ben-Dor et al., 2003), (8) **SAMBA**: statistical-algorithmic method for bicluster analysis (Tanay et al., 2002), (9) **xMOTIF**: conserved motifs (Murali & Kasif, 2003), (10) **Bimax**: divide-and-conquer algorithm (Prelic et al., 2006), (11) **CC**: Cheng-Church $\delta$-biclusters

(Cheng & Church, 2000), (12) **plaid_t**: improved plaid model (Turner et al., 2003), (13) **FLOC**: flexible overlapped biclustering, a generalization of **CC** (Yang et al., 2005), and (14) **spec**: spectral biclustering (Kluger et al., 2003).

For a fair comparison, the parameters of the methods were optimized on auxiliary toy data sets. If more than one setting was close to the optimum, all near optimal parameter settings were tested. In the following, these variants are denoted as *method_variant* (e.g. plaid_ss). For RFN we used the following parameter setting: 13 hidden units, a dropout rate of 0.1, 500 iterations with a learning rate of 0.1, and set the parameter $\alpha$ (controlling the sparseness on the weights) to 0.01.

## 3.2 SIMULATED DATA SETS WITH KNOWN BICLUSTERS

In the following subsections, we describe the data generation process and results for synthetically generated data according to either a multiplicative or additive model structure.

### 3.2.1 DATA WITH MULTIPLICATIVE BICLUSTERS

We assumed $n = 1000$ genes and $l = 100$ samples and implanted $p = 10$ multiplicative biclusters. The bicluster datasets with $p$ biclusters are generated by following model:

$$\boldsymbol{X} \ = \ \sum_{i=1}^{p} \boldsymbol{\lambda}_i \, \boldsymbol{z}_i^T \ + \ \boldsymbol{\Upsilon} \, , \tag{7}$$

where $\boldsymbol{\Upsilon} \in \mathbb{R}^{n \times l}$ is additive noise; $\boldsymbol{\lambda}_i \in \mathbb{R}^n$ and $\boldsymbol{z}_i \in \mathbb{R}^l$ are the bicluster membership vectors for the $i$-th bicluster. The $\boldsymbol{\lambda}_i$'s are generated by (i) randomly choosing the number $N_i^{\lambda}$ of genes in bicluster $i$ from $\{10, \dots, 210\}$, (ii) choosing $N_i^{\lambda}$ genes randomly from $\{1, \dots, 1000\}$, (iii) setting $\boldsymbol{\lambda}_i$ components not in bicluster $i$ to $\mathcal{N}(0, 0.2^2)$ random values, and (iv) setting $\boldsymbol{\lambda}_i$ components that are in bicluster $i$ to $\mathcal{N}(\pm 3, 1)$ random values, where the sign is chosen randomly for each gene. The $\boldsymbol{z}_i$'s are generated by (i) randomly choosing the number $N_i^{\boldsymbol{z}}$ of samples in bicluster $i$ from $\{5, \dots, 25\}$, (ii) choosing $N_i^{\boldsymbol{z}}$ samples randomly from $\{1, \dots, 100\}$, (iii) setting $\boldsymbol{z}_i$ components not in bicluster $i$ to $\mathcal{N}(0, 0.2^2)$ random values, and (iv) setting $\boldsymbol{z}_i$ components that are in bicluster $i$ to $\mathcal{N}(2, 1)$ random values. Finally, we draw the $\boldsymbol{\Upsilon}$ entries (additive noise on all entries) according to $\mathcal{N}(0, 3^2)$ and compute the data $\boldsymbol{X}$ according to Eq. (7). Using these settings, noisy biclusters of random sizes between $10 \times 5$ and $210 \times 25$ (genes$\times$samples) are generated. In all experiments, rows (genes) were standardized to mean 0 and variance 1.

### 3.2.2 DATA WITH ADDITIVE BICLUSTERS

In this experiment we generated biclustering data where biclusters stem from an additive two-way ANOVA model:

$$\boldsymbol{X} \ = \ \sum_{i=1}^{p} \boldsymbol{\theta}_i \odot (\boldsymbol{\lambda}_i \, \boldsymbol{z}_i^T) \ + \ \boldsymbol{\Upsilon} \quad , \quad \theta_{ikj} \ = \ \mu_i \ + \ \alpha_{ik} \ + \ \beta_{ij} \, , \tag{8}$$

where $\odot$ is the element-wise product of matrices and both $\boldsymbol{\lambda}_i$ and $\boldsymbol{z}_i$ are binary indicator vectors which indicate the rows and columns belonging to bicluster $i$. The $i$-th bicluster is described by an ANOVA model with mean $\mu_i$, $k$-th row effect $\alpha_{ik}$ (first factor of the ANOVA model), and $j$-th column effect $\beta_{ij}$ (second factor of the ANOVA model). The ANOVA model does not have interaction effects. While the ANOVA model is described for the whole data matrix, only the effects on rows and columns belonging to the bicluster are used in data generation. Noise and bicluster sizes are generated as in previous Subsection 3.2.1.

Data was generated for three different signal-to-noise ratios which are determined by distribution from which $\mu_i$ is chosen: A1 (low signal) $\mathcal{N}(0, 2^2)$, A2 (moderate signal) $\mathcal{N}(\pm 2, 0.5^2)$, and A3 (high signal) $\mathcal{N}(\pm 4, 0.5^2)$, where the sign of the mean is randomly chosen. The row effects $\alpha_{ki}$ are chosen from $\mathcal{N}(0.5, 0.2^2)$ and the column effects $\beta_{ij}$ are chosen from $\mathcal{N}(1, 0.5^2)$.

### 3.2.3 RESULTS ON SIMULATED DATA SETS

For method evaluation, we use the previously introduced biclustering consensus score for two sets of biclusters (Hochreiter et al., 2010), which is computed as follows:

Table 1: Results are the mean of 100 instances for each simulated data sets. Data sets M1 and A1-A3 were multiplicative and additive bicluster, respectively. The numbers denote average consensus scores with the true biclusters together with their standard deviations in parentheses. The best results are printed bold and the second best in italics ("better" means significantly better according to both a paired $t$-test and a McNemar test of correct elements in biclusters).

| Method | multiplic. model M1 | additive model A1 | A2 | A3 |
|---|---|---|---|---|
| RFN | **0.643±7e-4** | **0.475±9e-4** | **0.640±1e-2** | **0.816±6e-7** |
| FABIA | 0.478±1e-2 | 0.109±6e-2 | 0.196±8e-2 | 0.475±1e-1 |
| FABIAS | *0.564±3e-3* | *0.150±7e-2* | *0.268±7e-2* | *0.546±1e-1* |
| SAMBA | 0.006±5e-5 | 0.002±6e-4 | 0.002±5e-4 | 0.003±8e-4 |
| xMOTIF | 0.002±6e-5 | 0.002±4e-4 | 0.002±4e-4 | 0.001±4e-4 |
| MFSC | 0.057±2e-3 | 0.000±0e-0 | 0.000±0e-0 | 0.000±0e-0 |
| Bimax | 0.004±2e-4 | 0.009±8e-3 | 0.010±9e-3 | 0.014±1e-2 |
| plaid_ss | 0.045±9e-4 | 0.039±2e-2 | 0.041±1e-2 | 0.074±3e-2 |
| CC | 0.001±7e-6 | 4e-4±3e-4 | 3e-4±2e-4 | 1e-4±1e-4 |
| plaid_ms | 0.072±4e-4 | 0.064±3e-2 | 0.072±2e-2 | 0.112±3e-2 |
| plaid_t_ab | 0.046±5e-3 | 0.021±2e-2 | 0.005±6e-3 | 0.022±2e-2 |
| plaid_ms_5 | 0.083±6e-4 | 0.098±4e-2 | 0.143±4e-2 | 0.221±5e-2 |
| plaid_t_a | 0.037±4e-3 | 0.039±3e-2 | 0.010±9e-3 | 0.051±4e-2 |
| FLOC | 0.006±3e-5 | 0.005±9e-4 | 0.005±1e-3 | 0.003±9e-4 |
| ISA | 0.333±5e-2 | 0.039±4e-2 | 0.033±2e-2 | 0.140±7e-2 |
| spec | 0.032±5e-4 | 0.000±0e-0 | 0.000±0e-0 | 0.000±0e-0 |
| OPSM | 0.012±1e-4 | 0.007±2e-3 | 0.007±2e-3 | 0.008±2e-3 |

1. Compute similarities between all pairs of biclusters by the Jaccard index, where one is from the first set and the other from the second set;

2. Assign the biclusters of one set to biclusters of the other set by maximizing the assignment by the Munkres algorithm;

3. Divide the sum of similarities of the assigned biclusters by the number of biclusters of the larger set.

Step (3) penalizes different numbers of biclusters in the sets. The highest consensus score is 1 and only obtained for identical sets of biclusters.

Table 1 shows the biclustering results for these data sets. RFN significantly outperformed all other methods ($t$-test and McNemar test of correct elements in biclusters).

### 3.3 Gene Expression Data Sets

In this experiment, we test the biclustering methods on gene expression data sets, where the biclusters are gene modules. The genes that are in a particular gene module belong to the according bicluster and samples for which the gene module is activated belong to the bicluster. We consider three gene expression data sets which have been provided by the Broad Institute and were previously clustered by Hoshida et al. (2007) using additional data sets. Our goal was to study how well biclustering methods are able to recover these clusters without any additional information.

**(A)** The *"breast cancer" data set* (van't Veer et al., 2002) was aimed at a predictive gene signature for the outcome of a breast cancer therapy. We removed the outlier array S54 which leads to a data set with 97 samples and 1213 genes. In Hoshida et al. (2007), three biologically meaningful subclasses were found that should be re-identified.

**(B)** The *"multiple tissue types" data set* (Su et al., 2002) are gene expression profiles from human cancer samples from diverse tissues and cell lines. The data set contains 102 samples with 5565 genes. Biclustering should be able to re-identify the tissue types.

**(C)** The *"diffuse large-B-cell lymphoma (DLBCL)" data set* (Rosenwald et al., 2002) was aimed at predicting the survival after chemotherapy. It contains 180 samples and 661 genes. The three classes found by Hoshida et al. (2007) should be re-identified.

Table 2: Results on the (A) breast cancer, (B) multiple tissue samples, (C) diffuse large-B-cell lymphoma (DLBCL) data sets measured by the consensus score. An "nc" entry means that the method did not converge for this data set. The best results are in bold and the second best in italics ("better" means significantly better according to a McNemar test of correct samples in clusters). The columns "#bc", "#g", "#s" provide the numbers of biclusters, their average numbers of genes, and their average numbers of samples, respectively. RFN is two times the best method and once on second place.

| method | (A) breast cancer | | | | (B) multiple tissues | | | | (C) DLBCL | | | |
|---|---|---|---|---|---|---|---|---|---|---|---|---|
| | score | #bc | #g | #s | score | #bc | #g | #s | score | #bc | #g | #s |
| RFN | **0.57** | 3 | 73 | 31 | **0.77** | 5 | 75 | 33 | *0.35* | 2 | 59 | 72 |
| FABIA | *0.52* | 3 | 92 | 31 | 0.53 | 5 | 356 | 29 | **0.37** | 2 | 59 | 62 |
| FABIAS | *0.52* | 3 | 144 | 32 | 0.44 | 5 | 435 | 30 | *0.35* | 2 | 104 | 60 |
| MFSC | 0.17 | 5 | 87 | 24 | 0.31 | 5 | 431 | 24 | 0.18 | 5 | 50 | 42 |
| plaid_ss | 0.39 | 5 | 500 | 38 | *0.56* | 5 | 1903 | 35 | 0.30 | 5 | 339 | 72 |
| plaid_ms | 0.39 | 5 | 175 | 38 | 0.50 | 5 | 571 | 42 | 0.28 | 5 | 143 | 63 |
| plaid_ms_5 | 0.29 | 5 | 56 | 29 | 0.23 | 5 | 71 | 26 | 0.21 | 5 | 68 | 47 |
| ISA_1 | 0.03 | 25 | 55 | 4 | 0.05 | 29 | 230 | 6 | 0.01 | 56 | 26 | 8 |
| OPSM | 0.04 | 12 | 172 | 8 | 0.04 | 19 | 643 | 12 | 0.03 | 6 | 162 | 4 |
| SAMBA | 0.02 | 38 | 37 | 7 | 0.03 | 59 | 53 | 8 | 0.02 | 38 | 19 | 15 |
| xMOTIF | 0.07 | 5 | 61 | 6 | 0.11 | 5 | 628 | 6 | 0.05 | 5 | 9 | 9 |
| Bimax | 0.01 | 1 | 1213 | 97 | 0.10 | 4 | 35 | 5 | 0.07 | 5 | 73 | 5 |
| CC | 0.11 | 5 | 12 | 12 | nc | nc | nc | nc | 0.05 | 5 | 10 | 10 |
| plaid_t_ab | 0.24 | 2 | 40 | 23 | 0.38 | 5 | 255 | 22 | 0.17 | 1 | 3 | 44 |
| plaid_t_a | 0.23 | 2 | 24 | 20 | 0.39 | 5 | 274 | 24 | 0.11 | 3 | 6 | 24 |
| spec | 0.12 | 13 | 198 | 28 | 0.37 | 5 | 395 | 20 | 0.05 | 28 | 133 | 32 |
| FLOC | 0.04 | 5 | 343 | 5 | nc | nc | nc | nc | 0.03 | 5 | 167 | 5 |

For methods assuming a fixed number of biclusters, we chose five biclusters — slightly higher than the number of known clusters to avoid biases towards prior knowledge about the number of actual clusters. Besides the number of hidden units (biclusters) we used the same parameters as described in Sec. 3.1. The performance was assessed by comparing known classes of samples in the data sets with the sample sets identified by biclustering using the consensus score defined in Subsection 3.2.3 — here the score is evaluated for sample clusters instead of biclusters. The biclustering results are summarized in Table 2. RFN biclustering yielded in two out of three datasets significantly better results than all other methods and was on second place for the third dataset (significantly according to a McNemar test of correct samples in clusters).

## 3.4 1000 GENOMES DATA SETS

In this experiment, we used RFN for detecting DNA segments that are identical by descent (IBD). A DNA segment is IBD in two or more individuals, if they have inherited it from a common ancestor, that is, the segment has the same ancestral origin in these individuals. Biclustering is well-suited to detect such IBD segments in a genotype matrix (Hochreiter, 2013; Povysil & Hochreiter, 2014), which has individuals as row elements and genomic structural variations (SNVs) as column elements. Entries in the genotype matrix usually count how often the minor allele of a particular SNV is present in a particular individual. Individuals that share an IBD segment are similar to each other because they also share minor alleles of SNVs (tagSNVs) within the IBD segment. Individuals that share an IBD segment represent a bicluster.

For our IBD-analysis we used the next generation sequencing data from the 1000 Genomes Phase 3. This data set consists of low-coverage whole genome sequences from 2,504 individuals of the main continental population groups (Africans (AFR), Asians (ASN), Europeans (EUR), and Admixed Americans (AMR)). Individuals that showed cryptic first degree relatedness to others were removed, so that the final data set consisted of 2,493 individuals. Furthermore, we also included archaic human and human ancestor genomes, in order to gain insights into the genetic relationships between humans, Neandertals and Denisovans. The common ancestor genome was reconstructed from human, chimpanzee, gorilla, orang-utan, macaque, and marmoset genomes. RFN IBD detec-

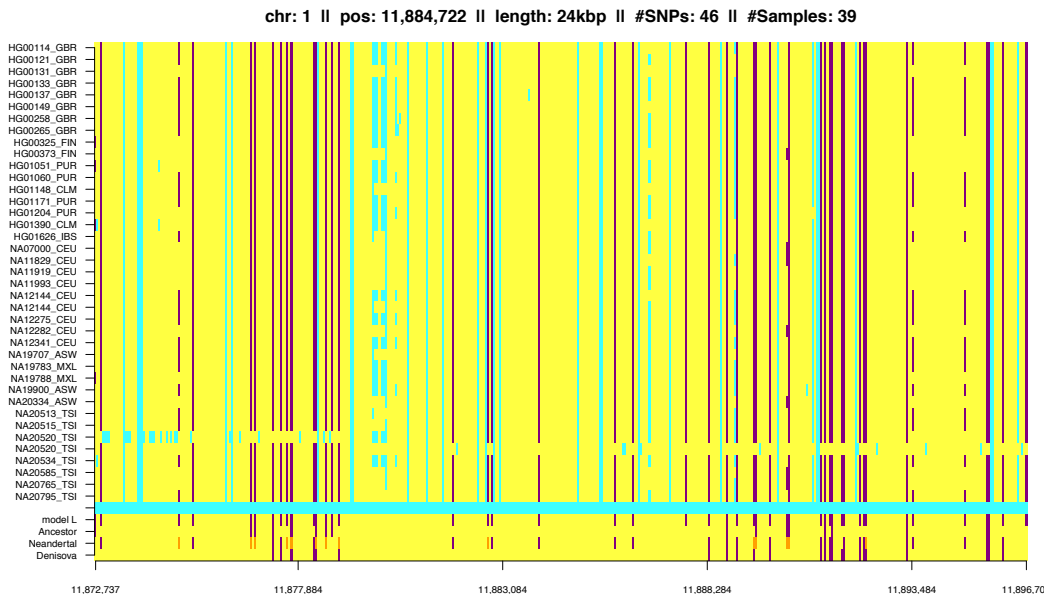

Figure 2: Example of an IBD segment matching the Neandertal genome shared among multiple populations. The rows give all individuals that contain the IBD segment and columns consecutive SNVs. Major alleles are shown in yellow, minor alleles of tagSNVs in violet, and minor alleles of other SNVs in cyan. The row labeled *model L* indicates tagSNVs identified by RFN in violet. The rows *Ancestor*, *Neandertal*, and *Denisova* show bases of the respective genomes in violet if they match the minor allele of the tagSNVs (in yellow otherwise). For the *Ancestor genome* we used the reconstructed common ancestor sequence that was provided as part of the 1000 Genomes Project data.

tion is based on low frequency and rare variants, therefore we removed common and private variants prior to the analysis. Afterwards, all chromosomes were divided into intervals of 10,000 variants with adjacent intervals overlapping by 5,000 variants

In the data of the 1000 Genomes Project, we found IBD-based indications of interbreeding between ancestors of humans and other ancient hominins within Africa (see Fig. 2 as an example of an IBD segment that matches the Neandertal genome).

## 4 CONCLUSION

We have introduced rectified factor networks (RFNs) for biclustering and benchmarked it with 13 other biclustering methods on artificial and real-world data sets.

On 400 benchmark data sets with artificially implanted biclusters, RFN significantly outperformed all other biclustering competitors including FABIA. On three gene expression data sets with previously verified ground-truth, RFN biclustering yielded twice significantly better results than all other methods and was once the second best performing method. On data of the 1000 Genomes Project, RFN could identify IBD segments which support the hypothesis that interbreeding between ancestors of humans and other ancient hominins already have taken place in Africa.

RFN biclustering is geared to large data sets, sparse coding, many coding units, and distinct membership assignment. Thereby RFN biclustering overcomes the shortcomings of FABIA and has the potential to become the new state of the art biclustering algorithm.

**Acknowledgment.** We thank the NVIDIA Corporation for supporting this research with several Titan X GPUs.

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
