# Peer review of "Rectified Factor Networks for Biclustering"

_ICLR 2017 — rejected_

[Official Review · AnonReviewer2 · rating 5 · confidence 2 · 08 Dec 2016]
**Review on the paper**

This paper applies RFN for biclustering to overcome the drawbacks in FABIA. The proposed method performs best among 14 biclustering methods, However, my first concern is that from the methodological point of view, the novelty of the proposed method seems small. The authors replied to the same question which another reviewer gave, but the replies were not so convincing. 

This paper was actually difficult for me to follow. For instance, in Figure 1, a bicluster matrix is constructed as an outer product of $h$ and $w$. $h$ is a hidden unit, but what is $w$? I could not find any definition. Furthermore, I could not know how $h$ is estimated in this method. Therefore, I do NOT understand how this method performs biclustering. 

Totally, I am not sure that this paper is suitable for publication. 

Prons:
Empirical performance is good.

Cons:
Novelty of the proposed method
Some description in the paper is unclear.

[Official Review · AnonReviewer3 · rating 5 · confidence 2 · 12 Dec 2016]
**Interesting paper, but the exact model being used is difficult to understand.**

Clarity: The novel contribution of the paper --- Section 2.2 --- was very difficult to understand. The notation seemed inconsistent (particularly the use of l, p, and m), and I am still not confident that I understand the model being used.

Originality: The novelty comes from applying the RFN model (including the ReLU non-linearity and dropout training) to the problem of biclustering. It sounds like a good idea. 

Significance: The proposed algorithm appears to be a useful tool for unsupervised data modelling, and the authors make a convincing argument that it is significant. (I.E. The previous state-of-the-art, FABIA, is widely used and this method both outperforms and addresses some of the practical difficulties with that method.)

Quality: The experiments are high-quality. 

Comments:
1) The introduction claims that this method is much faster than FABIA because the use of rectified units allow it to be run on GPUs. It is not clear to me how this works. How many biclusters can be supported with this method? It looks like the number of biclusters used for this method in the experiments is only 3-5?
2) The introduction claims that using dropout during training increases sparsity in the bicluster assignments. This seems like a reasonable hypothesis, but this claim should be supported with a better argument or experiments.
3) How is the model deep? The model isn't deep just because it uses a relu and dropout.

[Official Review · AnonReviewer1 · rating 4 · confidence 4 · 15 Dec 2016]
**Interesting work but poorly presented**

The paper presents a repurposing of rectified factor networks proposed
earlier by the same authors to biclustering. The method seems
potentially quite interesting but the paper has serious problems in
the presentation.


Quality:

The method relies mainly on techniques presented in a NIPS 2015 paper
by (mostly) the same authors. The experimental procedure should be
clarified further. The results (especially Table 2) seem to depend
critically upon the sparsity of the reported clusters, but the authors
do not explain in sufficient detail how the sparsity hyperparameter is
determined.


Clarity:

The style of writing is terrible and completely unacceptable as a
scientific publication. The text looks more like an industry white
paper or advertisement, not an objective scientific paper. A complete
rewrite would be needed before the paper can be considered for
publication. Specifically, all references to companies using your
methods must be deleted.

Additionally, Table 1 is essentially unreadable. I would recommend
using a figure or cleaning up the table by removing all engineering
notation and reporting numbers per 1000 so that e.g. "0.475 +/- 9e-4"
would become "475 +/- 0.9". In general figures would be preferred as a
primary means for presenting the results in text while tables can be
included as supplementary information.


Originality:

The novelty of the work appears limited: the method is mostly based on
a NIPS 2015 paper by the same authors. The experimental evaluation
appears at least partially novel, but for example the IBD detection is
very similar to Hochreiter (2013) but without any comparison.


Significance:

The authors' strongest claim is based on strong empirical performance
in their own benchmark problems. It is however unclear how useful this
would be to others as there is no code available and the details of
the implementation are less than complete. Furthermore, the method
depends on many specific tuning parameters whose tuning method is not
fully defined, leaving it unclear how to guarantee the generalisation
of the good performance.

[Final Decision · Program Chairs · 06 Feb 2017]
**ICLR committee final decision**

The reviewers pointed out several issues with the paper, and all recommended rejection.